# Mitochondrial Calcium Overload Plays a Causal Role in Oxidative Stress in the Failing Heart

**DOI:** 10.3390/biom13091409

**Published:** 2023-09-19

**Authors:** Haikel Dridi, Gaetano Santulli, Laith Bahlouli, Marco C. Miotto, Gunnar Weninger, Andrew R. Marks

**Affiliations:** 1Department of Physiology and Cellular Biophysics, Clyde and Helen Wu Center for Molecular Cardiology, Columbia University Vagelos College of Physicians & Surgeons, New York, NY 10032, USA; lbahlouli25@amherst.edu (L.B.); mm5642@cumc.columbia.edu (M.C.M.); gw2424@cumc.columbia.edu (G.W.); arm42@cumc.columbia.edu (A.R.M.); 2Department of Medicine, Division of Cardiology, Wilf Family Cardiovascular Research Institute, Albert Einstein College of Medicine, New York, NY 10461, USA; gsantulli001@gmail.com

**Keywords:** mitochondria, calcium, heart failure

## Abstract

Heart failure is a serious global health challenge, affecting more than 6.2 million people in the United States and is projected to reach over 8 million by 2030. Independent of etiology, failing hearts share common features, including defective calcium (Ca^2+^) handling, mitochondrial Ca^2+^ overload, and oxidative stress. In cardiomyocytes, Ca^2+^ not only regulates excitation–contraction coupling, but also mitochondrial metabolism and oxidative stress signaling, thereby controlling the function and actual destiny of the cell. Understanding the mechanisms of mitochondrial Ca^2+^ uptake and the molecular pathways involved in the regulation of increased mitochondrial Ca^2+^ influx is an ongoing challenge in order to identify novel therapeutic targets to alleviate the burden of heart failure. In this review, we discuss the mechanisms underlying altered mitochondrial Ca^2+^ handling in heart failure and the potential therapeutic strategies.

## 1. Introduction

Heart failure (HF) is a complex clinical syndrome characterized by the inability of the heart to pump blood at a rate and with adequate force to meet the metabolic demands of the body. This condition can result from various underlying causes, including damage to the heart muscle (such as from a heart attack), hypertension, valvular heart disease, or other cardiac conditions. HF can affect either the left, right, or both ventricles of the heart and can manifest as systolic dysfunction (impaired ability of the heart to contract and eject blood) or diastolic dysfunction (impaired ability of the heart to relax and fill with blood). The hallmark symptoms of HF include shortness of breath, fatigue, fluid retention (edema), and reduced exercise tolerance. It is a chronic condition that requires ongoing management and treatment to improve the patient’s quality of life and reduce morbidity and mortality. HF shows a steadily growing prevalence and remains the leading cause of death in developed countries [1,2]. Although initially adaptive [3,4,5], cardiac responses to pressure or volume overload are associated with deep molecular changes, eventually leading to fetal gene expression [6,7,8,9], impaired contractile function [10,11], abnormal vascularization [12,13], altered extracellular matrix composition [14,15,16,17], fibrosis [18,19], and profound metabolic abnormalities [20,21,22,23,24,25,26,27], all of which unavoidably affect myocardial contraction and eventually progress towards overt HF [28].

The heart consumes a significant amount of energy supplied almost entirely by mitochondrial ATP production derived from oxidative metabolism [29]. Nearly a third of each cardiomyocyte, by volume, is taken up by mitochondria [30], which are located in close proximity to the sarcoplasmic reticulum (SR), allowing the two organelles to exchange metabolites and interconnect [31,32,33,34]. Calcium (Ca^2+^) and reactive oxygen species (ROS) are generally considered the main transduction signals linking the SR and mitochondrion and permit them to adapt their responses to external stress in a highly regulated manner [35,36,37,38,39,40]. Independent of etiology, a key hallmark of HF is Ca^2+^ dyshomeostasis, altered bioenergetics, changes in mitochondrial respiration, increased oxidative stress, and impaired energy production and consumption [41,42,43,44]. During HF, impaired Ca^2+^ uptake due to dysregulation in several calcium cycling proteins and increased SR Ca^2+^ leak through type-2 ryanodine receptors (RyR2) resulting in reduced intracellular Ca^2+^ stores required for cardiomyocyte contraction and increased baseline cytosolic Ca^2+^ [45,46,47,48,49,50]. Increased mitochondrial Ca^2+^ uptake follows, which contributes to mitochondrial Ca^2+^ overload, dysfunction, and oxidative stress [51]. Mitochondrial dysfunction increases mitochondrial ROS production, which has secondary effects on a broad range of cellular functions and accelerates HF progression [52,53,54,55,56].

In this review, we summarize the recent evidence demonstrating the crucial role of mitochondrial Ca^2+^ overload in inducing oxidative stress during HF and highlight the potential therapeutic targets as well as new outlooks for future studies.

## 2. Mitochondrial Function in Cardiomyocytes

The heart primarily depends on mitochondrial oxidative metabolism to fulfill approximately 95% of its energy demands [29], with a particular reliance on ATP usage during the process of excitation–contraction (EC) coupling. Within the mitochondria, oxidative metabolism is chiefly propelled by redox reactions occurring in the electron transport chain (ETC). The reduced forms of nicotinamide adenine dinucleotide (NADH) and flavin adenine dinucleotide (FADH2) serve as electron donors, and these electrons traverse a series of increasingly reduced electron carriers, ultimately reaching oxygen, which serves as the final electron acceptor [57,58]. The energy generated from this exergonic flow of electrons is utilized to move protons from the mitochondrial matrix to the intermembrane space, thus establishing a chemical and electrical gradient across the inner mitochondrial membranes [59,60,61,62]. This gradient, impermeable to ions, represents the proton motive force, which is then harnessed by ATP synthase to convert ADP into ATP through oxidative phosphorylation [63]. A continuous flow of electrons along the ETC is sustained by the dehydrogenases found in the tricarboxylic acid cycle, also known as the Krebs cycle, which consistently supplies NADH and FADH2 [64]. In the intermembrane space, mitochondrial creatine kinase (CK) transforms creatine transported from the cytosol into phosphocreatine, serving as a spatial and temporal reservoir of high-energy phosphates [28,65].

Increased ATP consumption in cardiomyocytes requires additional reducing equivalents, and Ca^2+^ ions are essential to this process. Indeed, increased cytosolic Ca^2+^ levels with elevated cardiac contractions allow Ca^2+^ to enter the mitochondria and stimulate pyruvate dehydrogenase, α-ketoglutarate, and isocitrate dehydrogenase to regenerate reducing equivalents [66], as well as signal ATP synthase and complex III of the ETC to accelerate ATP production [67]. However, persistently high levels of Ca^2+^ entry into the mitochondria, as in HF, may be deleterious, leading to mitochondrial Ca^2+^ overload, increased ROS production, and oxidative stress [68]. The fundamental molecular mechanisms leading to mitochondrial Ca^2+^ overload during HF are discussed in the following sections.

## 3. Ca^2+^ Homeostasis Dysregulation and Mitochondrial Ca^2+^ Overload in HF

### 3.1. Cardiac Ca^2+^ Handling

Cardiac muscle contraction is regulated beat-to-beat by Ca^2+^ stored and released within cardiomyocytes in EC coupling. During the depolarizing phase of the cardiac action potential, Ca^2+^ enters the cardiomyocyte through voltage-activated L-type Ca^2+^ channels and triggers RyR2 to open and release Ca^2+^ from SR Ca^2+^ stores through calcium-induced calcium release (CICR), raising cytosolic Ca^2+^ levels about ten-fold to ~1 μM [46]. Ca^2+^ then activates specific proteins of the contractile apparatus and induces contraction of cardiac cells [49,69,70]. Subsequent relaxation occurs as Ca^2+^ is removed from the cytosol via three primary mechanisms: type-2a SR Ca^2+^-ATPase (SERCA2a) transport, sarcolemmal Na^+^/Ca^2+^ exchanger (NCX) extrusion, and mitochondrial Ca^2+^ uptake [71,72,73,74,75].

During HF, the widespread dysregulation of Ca^2+^ cycling leading to significant increases in cytosolic baseline Ca^2+^ has been observed [76]. First, the increased activation and open probabilities of voltage-gated L-type Ca^2+^ channels in failing compared with non-failing human ventricles was confirmed, which resulted in an excessive calcium influx into the cytosol [77,78]. Moreover, the hyperactivity of type-2 ryanodine receptors (RyR2) due to increased CICR and post-translational modification, which culminated in a significant SR Ca^2+^ leak, was also shown in human and mice cardiomyocytes of failing hearts [76]. Furthermore, a reduced effectiveness of the sarcolemmal Na^+^/Ca^2+^ exchanger (NCX) in extruding cytosolic calcium was demonstrated in a mouse model of heart failure [79]. Finally, reduced SERCA2a expression and activity was observed in several animal models of heart failure and was shown to decrease SR calcium uptake from the cytosol [80,81]. These defects result in increased cytosolic Ca^2+^ and potentially increase Ca^2+^ entry into the mitochondria, causing a detrimental mitochondrial Ca^2+^ overload in cardiomyocytes [82,83].

### 3.2. Inflammation-Mediated Ca^2+^ Dyshomeostasis in HF

Inflammation has emerged as a major pathophysiological feature of HF progression, regardless of etiology. Numerous pro-inflammatory signaling molecules are produced and released in HF, initiating vicious cascades involving oxidative stress, mitochondrial dysfunction, myocardial weakened contractility, and, importantly, Ca^2+^ dyshomeostasis [84,85]. Elevated levels pro-inflammatory cytokines, including tumor necrosis factor (TNF-⍺), interleukin-17 (IL-17), interleukin-6 (IL-6), interleukin-1 (IL-1ß), and transforming growth factor (TGF-ß), found in HF were all associated with the dysregulation of several important Ca^2+^ cycling proteins [84,85]. TNF-⍺ was found to be significantly elevated in the plasma of heart failure patients, serving as an effective predictor of HF severity and diastolic dysfunction [86,87]. Several studies have shown that TNF-⍺ suppresses SERCA2a gene expression, worsening cytosolic Ca^2+^ overload [86,87]. Increased levels of IL-17, IL-6, and IL-1ß in HF were all associated with left ventricular dysfunction, causing further SERCA2a gene downregulation [88,89]. Finally, TGF-ß was demonstrated to be significantly upregulated in HF, associated with cardiac injury, cardiomyocyte apoptosis, hypertrophy, and fibrosis [84,85]. Waning et al. report that this increased TGF-ß activity can lead to NADPH Oxidase 2 (NOX2) localization to RyR2, causing channel oxidation due to superoxide radicals (O_2_·^−^) and resulting in an SR Ca^2+^ leak [90]. The elevated release of these pro-inflammatory cytokines in HF would increase cytosolic baseline Ca^2+^ and mitochondrial Ca^2+^ overload, leading to ROS formation and oxidative stress. The signaling cascades associating the inflammatory response to SR/mitochondrial dysfunction in HF still remain to be elucidated.

### 3.3. Mitochondrial Ca^2+^ Handling

#### 3.3.1. Role of the Mitochondrial Ca^2+^ Uniporter

Mitochondrial Ca^2+^ influx occurs primarily via the mitochondrial Ca^2+^ uniporter (MCU), whereas Ca^2+^ efflux occurs mainly via the mitochondrial Na^+^-Ca^2+^—Li^+^ exchanger (mNCLX) [91,92,93,94]. MCU, previously known as the coiled-coil domain containing 109A (CCD109A) and C10 or f42, is a ruthenium-sensitive Ca^2+^ channel in the mitochondrial inner membrane that facilitates Ca^2+^ transport down its electrochemical gradient without coupling Ca^2+^ transport with the movement of other ions [95,96,97,98,99,100,101,102,103,104,105,106,107,108,109]. The mitochondrial Ca^2+^ uptake channel is a large holocomplex consisting of MCU as the pore-forming unit and multiple regulatory proteins, including the essential MCU regulator (EMRE) [110,111,112,113], and EF-hand proteins MICU1 [114,115,116,117,118,119,120,121,122,123,124], MICU2 [125,126,127,128], and MICU3 [127,128,129,130,131,132]. The study of the cardiac phenotype of murine models lacking MCU yielded controversial results. First and foremost, the germline mutation of MCU was embryonically lethal in the C57BL/6 background but not in the CD background. The latter mice showed modest differences in the myocardial contractile response to isoproterenol and did not seem to be protected from cardiac ischemia–reperfusion injury [103]. These results were further confirmed in a cardiac-specific dominant negative MCU mouse [133]. However, the cardiac-specific inducible MCU mouse exhibited a significant difference in the contractile response to adrenergic stimulation [134,135]; however, that finding was not reproduced when the hearts were studied in an isolated system [136].

During HF, increased cytosolic Ca^2+^ leads to excessive Ca^2+^ entry into the mitochondria and causes mitochondrial Ca^2+^ overload, which is detrimental to the cardiomyocyte [82,83,137]. Considering that excessive mitochondrial Ca^2+^ uptake causes mitochondrial Ca^2+^ overload and oxidative stress, MCU modulators can be an interesting therapeutic option to rescue the subsequent detrimental signaling cascades observed in HF [122,136,138,139,140,141,142,143,144,145,146]. Several studies have reported that the inhibition of mitochondrial Ca^2+^ uptake during acute stress can be reduced by the deletion of the MCU gene or by the MCU blocker, Ru360 [147,148], while basal mitochondrial Ca^2+^ levels remain unchanged [134,135]. Furthermore, MCU deletion was shown to not completely inhibit mitochondrial Ca^2+^ uptake [149]. These findings suggest the existence of other mechanisms of mitochondrial Ca^2+^ uptake with different levels of sensitivity to cytosolic Ca^2+^. Possible mediators of MCU-independent mitochondrial Ca^2+^ uptake include the Ca^2+^/H^+^ exchanger Leucine Zipper And EF-Hand Containing Transmembrane Protein 1 (Letm1) [150,151,152,153,154,155,156] and the Transient Receptor Potential Canonical 3 (TRPC3) [130,157,158,159,160,161,162,163]. These aspects are not surprising since MCU Ca^2+^ sensitivity has been shown [164] to be quite low (K_d_~20–30 μMol) and, as such, resting cytosolic Ca^2+^ can be insufficient to trigger MCU activity [120,139,165,166]. On the other hand, it is important to define what the basal cytosolic Ca^2+^ concentration in cardiomyocytes is, as well as the difference between basal cytosolic Ca^2+^ in normal versus failing cardiomyocytes when associated with a defective RyR2 leak or reduced SERCA2a activity. Nonetheless, is basal cytosolic Ca^2+^ in HF sufficient to enter mitochondria and cause Ca^2+^ overload? This dilemma represents an important question that needs to be clarified by future investigations.

#### 3.3.2. Role of the mNCLX

Recently, Luongo and collaborators revealed a novel mechanism, independent of MCU, responsible for mitochondrial Ca^2+^ homeostasis [167]. Since mitochondrial Ca^2+^ uptake has to be balanced with extrusion in homeostatic conditions, they hypothesized that the key channel responsible for mitochondrial efflux Ca^2+^, mNCLX [94,167], would play a crucial role in mitochondrial Ca^2+^ handling [167]. Using a murine model harboring a tamoxifen-induced deletion of mNCLX, they demonstrated that these mice developed left ventricular dilation and decreased cardiac function. This was mirrored by mitochondrial remodeling and dysfunction followed by death within 2 weeks of tamoxifen delivery. On the other hand, mice overexpressing mitochondrial mNCLX, which enhanced mitochondrial Ca^2+^ efflux by 38%, displayed an attenuated infarct size after MI or I/R injury, improved cardiac function, and reduced superoxide generation [167]. Despite the promising outcomes of mNCLX overexpression, whether mitochondria Ca^2+^ overload in HF is due to enhanced Ca^2+^ uptake or repressed extrusion has yet to be further elucidated. Furthermore, the following queries need to be investigated: does increased mNCLX expression/activity affect cytosolic Ca^2+^ levels by increasing mitochondrial Ca^2+^ extrusion? Moreover, what are the long-term outcomes of enhanced mNCLX activity in mice? Will those mice be more vulnerable to arrhythmias due to cytosolic Ca^2+^ overload? We believe that the long-term outcomes of this enhanced mNCLX activity are deteriorated when cardiac disease is associated with leaky RyR2 channels. Although, these hypotheses remain to be verified, recent studies by Garbincius point towards the deleterious effect of mNCLX cardiomyocyte-specific overexpression in mice. Indeed, survival was reduced when mNCLX cardiomyocyte-specific overexpressing mice were subjected to severe neurohormonal stress with angiotensin II and phenylephrine [168].

#### 3.3.3. Factors influencing Mitochondrial Ca^2+^ Uptake

It is well established that MCU does not transport Ca^2+^ when the Ca^2+^ concentration is below a threshold of 200 nM [169]. However, such a low affinity might be overcome when MCU is juxtaposed with the SR Ca^2+^ release channels RyR2 and/or 1,4,5-trisphosphate receptors (IP3Rs) [51,170,171]. Remarkably, hyper-physiological Ca^2+^ concentrations (10–100 µM) are required to activate Ca^2+^ uptake into isolated mitochondria; however, in intact cardiomyocytes, the cytosolic Ca^2+^ concentration after Ca^2+^ release from the SR remains less than 10 µM, yet Ca^2+^ is still internalized by mitochondria [172,173].

Such a discrepancy between isolated mitochondria and intact cells was partially resolved by the finding of high cytosolic Ca^2+^ concentrations in microdomains between mitochondria and the SR. For example, RyR2 Ca^2+^ release during systole creates microdomains of high, localized Ca^2+^ concentrations (~30 µM) in the vicinity of the mitochondrial membrane, leading to mitochondrial Ca^2+^ uptake [174,175,176,177,178,179,180,181,182,183]. This is an exciting theory; however, it is missing important pieces of the SR–mitochondria Ca^2+^ transfer puzzle that mostly concern the complexity of this highly regulated process, which involves two dynamic organelles. Due, in part, to the low Ca^2+^ affinity of the MCU, we believe that the formation of Ca^2+^-rich microdomains alone is not sufficient to allow Ca^2+^ to enter the mitochondria, especially in the presence of competing sarcolemmal NCX and SERCA2a pumps with a higher Ca^2+^ affinity and faster uptake rate.

Thus, a very close proximity between the two organelles seems to be necessary to allow Ca^2+^ to transfer to the mitochondria. This hypothesis was recently investigated and two major, non-mutually exclusive interactions between the SR and mitochondria were identified. The functional, tight coupling between the SR and mitochondria is attributed to the inter-organelle tether protein involving mitofusin2 (MFN2) [184,185,186] and/or the IP3R-Grp75-VDAC protein complex [33,143,187,188]. The relevance of each theory to the mitochondrial Ca^2+^ overload observed in HF is discussed below.

## 4. Physical SR–Mitochondria Interaction

### 4.1. Mitofusin-Mediated Tethering

Recently, the purification of subcellular fractions corresponding to endoplasmic reticulum (ER)–mitochondria contacts, referred to as mitochondria-associated membranes (MAMs), led to the identification of proteins enriched in those membrane domains [189,190,191]. Among them, MFN2 is located on both the ER and mitochondrial membranes [192,193], where it forms homo- and heterotypic interactions with mitofusin1 (MFN1) [194]. In addition, a relevant fraction of MFN2 (7–10%) exists in the ER and its ablation alters the structure of this organelle, causing its fragmentation [195,196]. The selective reconstitution of the ER pool of MFN2 in the same cells completely restores the reticular nature of the ER [184], suggesting an essential role of MFN2 in the modulation of ER shape.

MFN2-KO cells were used to investigate how tethering between the ER and mitochondria would impact Ca^2+^ transfer to the mitochondria. The lack of MFN2 to tether the ER to mitochondria resulted in a significant reduction in mitochondrial Ca^2+^ uptake, without any impairment of mitochondrial capacity to uptake Ca^2+^. This impaired uptake reflects an increased distance between the two organelles and, therefore, a limited generation of Ca^2+^ microdomains. Therefore, MFN2-KO cells provide experimental proof for the importance of Ca^2+^ microdomains between the ER/SR and mitochondria, a theory postulated by Rizzuto and Pozzan almost 30 years ago [175,197,198]. Thus, the physical juxtaposition of the SR to mitochondria by MFN2 appears to be essential for normal inter-organelle Ca^2+^ signaling in the myocardium. This is consistent with a requirement for SR–mitochondrial Ca^2+^ signaling through microdomains to maintain a competent bioenergetic feedback response in cardiomyocytes [192].

Interestingly, quantitative analyses have shown an increase, not a decrease, in the close contacts between the ER and mitochondria in MFN2-KO mouse embryonic fibroblasts, as compared with WT cells [199,200]. As previously described, the tethering of the SR to mitochondria in cardiac cells would be expected to facilitate the exchange of Ca^2+^ signals through inter-organelle microdomains, where the signaling factor is prevented from diffusing into the cytosol. However, as mitochondria are the main source of ATP in cardiomyocytes and the SR is the main source of Ca^2+^, it is reasonable to postulate the existence of a physical, dynamic interaction that facilitates direct Ca^2+^ exchange between the two organelles. Increased beat-to-beat SR Ca^2+^ release can then be rapidly sensed by the mitochondria, inducing mitochondrial Ca^2+^ uptake, stimulating mitochondrial dehydrogenases, increasing the production of NADH for the ETC, and increasing ATP production according to the physiological demand (see Figure 1). Furthermore, mitochondrial ATP would then be available to the SR to drive energy-intensive and phasic Ca^2+^ cycling. Nevertheless, the relevance of this finding for cardiomyocytes experiencing HF continues to be debated.

### 4.2. IP3R-Grp75-VDAC Complex and Mitochondrial Ca^2+^ Uptake

In the same vein as the close proximity between the SR and mitochondria described above, recent studies have identified several proteins enriched at the SR–mitochondria interface, highlighting the emerging role of inter-organelle interactions within the cell [184,201]. One important protein complex composed of the voltage-dependent ion channel (VDAC), chaperone glucose-regulated protein 75 (Grp75), and IP3R1 has been identified as a key regulator of direct Ca^2+^ transfer from the ER/SR to the mitochondria [33]. Specifically, VDAC1 was shown to be physically linked to IP3R1 through the chaperone protein Grp75 [192]. The functional interaction between these channels and the mitochondrion was verified by the recombinant expression of only the ligand-binding domain (LBD) of IP3R1 on the ER. Without any IP3R1 Ca^2+^ channel functionality, the LBD was able to directly enhance Ca^2+^ accumulation in the mitochondria through the outer mitochondrial membrane. Knocking down Grp75 abolished this stimulatory effect, highlighting the importance of chaperone-mediated conformational coupling between IP3R and the mitochondrial Ca^2+^ uptake machinery [33]. This finding was recently confirmed by other methods, including an in situ proximity ligation assay [202]. However, most studies, including ours, indicated that the activation of IP3Rs in cardiomyocytes, unlike RyR2, did not significantly contribute to EC coupling [51,203]. Furthermore, the relevance of IP3R to mitochondrial Ca^2+^ overload in HF remains mostly unclear. The removal of IP3R2 channels from cardiomyocytes did not have any impact on the progression of heart failure or on factors, such as Ca^2+^ sparks, SR Ca^2+^ load, mitochondrial Ca^2+^ levels, or the production of reactive oxygen species (ROS). This suggests that these channels do not play a role in the malfunctioning of Ca^2+^ regulation in ventricular cardiomyocytes [51]. Interestingly, when IP3R1 was eliminated, it led to a reduction in vasoconstriction in the coronary arteries, which was mediated by the activation of vascular smooth muscle cells, and consequently, it slowed down the advancement of HF [202]. On another note, our recent research demonstrated a strong association between mitochondrial Ca^2+^ overload in HF and the leakage of Ca^2+^ from RyR2 channels [51]. This discovery was further validated using mouse models with persistently leaky RyR2 channels, which displayed changes in the mitochondrial structure, a decrease in ATP content, and an increase in the production of mitochondrial ROS [51].

Despite the important progress achieved in the investigation of the pathophysiologic implications of mitochondrial Ca^2+^ overload, the precise quantity of Ca^2+^ taken up by mitochondria with each heartbeat in both normal and failing hearts remains uncertain. Furthermore, there is an ongoing debate regarding whether the mitochondrion can function as a primary reservoir for Ca^2+^ ions [204,205,206,207]. However, numerous studies have used genetic methods to demonstrate substantial mitochondrial dysfunction when Ca^2+^ homeostasis is manipulated [110,111]. Indeed, an excess of cytosolic Ca^2+^ in various animal models has led to a mitochondrial phenotype resembling that observed in heart failure. This phenotype includes compromised ATP generation, heightened mitochondrial oxidative stress, and, ultimately, the death of cardiomyocytes [208,209].

Other proteins facilitating a physical connection between the ER and mitochondria have been proposed, including PDZ-domain-containing 8 (PDZD8) [210,211], protrudin [212], Rab7 [213], FUN14-domain-containing 1 (FUNDC1), and protein tyrosine phosphatase-interacting protein 51 (PTPIP51) [214]. Christopher Miller’s lab demonstrated the interaction between vesicle-associated membrane protein-associated protein B (VAPB), expressed on the ER, and PTPIP51, located on the outer mitochondrial membrane [215]. The disruption of this interaction was shown to profoundly alter inter-organelle Ca^2+^ fluxes between the ER and mitochondria [216]; however, the exact role of RyR Ca^2+^ leak was not investigated in these models.

## 5. Mitochondrial Ca^2+^ Overload and Mitochondrial ROS Production

Numerous studies have shown that excessive mitochondrial Ca^2+^ levels lead to the production of mitochondrial reactive oxygen species in various diseases, including heart failure [51,68]. ROS are chemically reactive species based on oxygen (O2) that can rapidly interact with cellular components, such as lipids [217], proteins [218], and nucleic acids [219]. This category encompasses highly reactive molecules, including free radicals, such as superoxide (O_2_·^−^) and hydroxyl radical (·OH), as well as non-radical species, such as hydrogen peroxide (H_2_O_2_) and hypochlorite (OCl^−^) [220,221]. In mitochondria, the presence of Ca^2+^ stimulates the Krebs cycle, resulting in the generation of NADH and FADH2, both of which provide electrons to the electron transport chain (ETC). As electrons progress through a series of redox reactions along the ETC, a proton gradient is established, facilitating ATP production, as described earlier [222]. Depending on the cellular respiratory state, as much as 20% of the electrons within the ETC may experience a leakage to molecular oxygen to form superoxide anions [223,224,225]. In the mitochondria, much of the superoxide anions are dismutated to hydrogen peroxide (H_2_O_2_) by superoxide dismutase (SOD), with SOD2 being the primary mitochondrial isoform [226,227]. Mitochondrial H_2_O_2_ is then removed by the antioxidant systems of peroxiredoxin (Prx) and glutathione peroxidase (Gpx) in the mitochondria [228,229]. In HF, increased O_2_·^−^ generation from complex I of the ETC caused excessive formation of H_2_O_2_ and ·OH, which became deleterious when their production overwhelmed detoxification [230,231]. Since several ion channels are regulated by redox reactions, oxidative stress can further impact EC coupling either by directly targeting ion channels involved in EC coupling or by activating stress kinases, which may contribute to contractile dysfunction and the progression of HF [49,232]. In the following section, we focus on the key proteins involved in EC coupling that are targeted by mitochondrial ROS production and causative of increased mitochondrial Ca^2+^ overload. This overload further amplifies mitochondrial oxidative stress, propagating a vicious cycle.

## 6. Redox-Regulated Proteins of the Ca^2+^ Handling Machinery and HF

### 6.1. Ryanodine-Receptor Ca^2+^ Release Channel

Increased mitochondrial oxidative stress during HF, in human patients as well as in animal models [233,234], is commonly associated with an SR Ca^2+^ leak through the hyperactivity of RyR2 [46]. As HF progresses, oxidative stress worsens due to the increasing energy demand and workload of the failing heart [229]. Increased RyR2 oxidation and Ca^2+^ leak not only augments cytosolic Ca^2+^ concentrations by depleting Ca^2+^ stores but also increases mitochondria Ca^2+^ overload, which further produces ROS [51] and worsens the prognosis of HF. A deleterious cycle is therefore perpetuated.

Many cysteine residues in RyR can be subjected to oxidative post-translational modification; however, the exact nature of these modifications for available cysteines and how they affect the activity of the channel remains to be determined. Recent work suggests that, in the presence of H_2_O_2_, RyR undergoes covalent disulfide cross-linking between the N-terminal domains of the four neighboring protomers, adopting an open state conformation and generating an SR Ca^2+^ leak [235]. We and others have published extensive studies showing that the oxidation of the channel dissociates the stabilizing subunit, calstabin2, which results in increased activity and Ca^2+^ leak through the channel [50,51,236,237]. As such, the oxidation of RyR2 in HF is far from a simple signaling pathway and needs extensive research. Identifying the oxidized residues following HF, as well as the responsible ROS species, would enhance our understanding of RyR2 oxidation-mediated HF. Recent advances in resolving the structure of RyR2 in HF at a high resolution [45,238,239,240,241] using single-particle cryogenic-electron microscopy (cryo-EM) would be the perfect tool for solving the complete structure of RyR at an atomic resolution in order to clarify the architecture of multiple functional states, ligand-binding sites, and gating mechanisms, including RyR2 abnormalities, during HF. In the following section, we provide structural insights and future directions for the investigation of the redox-sensitive residues within RyR2 channels.

#### Potentially Oxidized Residues in RyR2

RyR channels comprise several surface-exposed cysteine residues with free thiol groups prone to oxidation by reactive oxygen and nitrogen species. Oxidized cysteines that render RyR2 channels leaky to Ca^2+^ have been implicated as a key factor for the development of age-related cardiac disorders, HF, and atrial fibrillation [157]. Assessing the cellular redox state of these cysteine residues is experimentally challenging because free thiol groups exhibit a high sensitivity to the redox potential of the experimental environment and may be altered accordingly (they can be oxidized or reduced during tissue isolation, preparation, storage, experiments, etc.). This poses an experimental limitation in defining a specific target or mechanism of action for RyR oxidation. Using a maleimide derivative to mask and label reactive cysteines, Voss et al. identified seven hyper-reactive cysteines (rabbit: C1040, C1303, C2436, C2565, C2606, C2611, and C3635) in RyR1, the skeletal muscle isoform [242]. On the other hand, Aracena-Parks et al. identified 12 hyper-active cysteines [243] in RyR1 (rabbit: C36, C253, C315, C811, C906, C1040, C1303, C1591, C2326, C3193, and C3635), three of which overlapped with those identified by Voss et al. (i.e., C1040, C1303, and C3635). These hyper-active cysteines are evolutionarily conserved in RyR1 across mammalian species and 9 of those are also found to be conserved in the RyR2 isoform (human: C36, C822, C917, C1582, C2402, C2572, C2577, C3158, and C3602). Although the functional significance of these oxygen reactive cysteines in RyR1 remains largely unknown, both studies identify cysteine RyR1-C3635/RyR2-C3602 as the most hyper-reactive residue. This particularly important residue is located in the calmodulin-binding domain (see Figure 2). These studies raised an important question regarding the functional significance of the corresponding RyR2 cysteine residue that is evolutionarily conserved in humans, C3602. Subsequently, Mi et al. found that the mutation of cysteine to alanine (C3602A) suppressed the store-overload-induced Ca^2+^ release (SOICR) by raising the activation threshold and delaying the termination of Ca^2+^ release. Furthermore, the C3602A mutation markedly increased fractional Ca^2+^ release and diminished the inhibitory effect of N-ethylmaleimide on Ca^2+^ release, but had no effect on the stimulatory action of 4,4′-dithiodipyridine on Ca^2+^ release [244]. Moreover, the C3602A mutation did not abolish the effect of calmodulin on Ca^2+^ release. Likewise, Nikolaienko et al. reported that this particular residue did not participate in inter-subunit cross-linking and played a limited role in RyR2 regulation by calmodulin during oxidative stress [245]. Based on our experience with high-resolution structures of RyR1 [246] and our ongoing research on the structure of human RyR2 [247], we mapped these previously identified residues to the 3D structure of RyR2 and discussed their potential relevance to RyR2 function (Figure 2). The 3D mapping confirmed that C3602 was located in the shell-core linker at the interface of the calmodulin binding site.

The other hyper-reactive cysteines C36, C253, and C315 are located in the N-terminal domain (NTD) of RyR1, while only C36 is conserved in RyR2. The N-terminal β-trefoil domains form the central cytosolic vestibule mediating inter-subunit interactions in the resting state of the channel. Since these interactions are disrupted during channel activation, it has been proposed that NTD is involved in the allosteric regulation of channel gating in both RyR1 and RyR2. Importantly, NTD is a known disease hot spot for pathogenic CPVT mutations weakening the RyR2 stability of the closed state [248,249,250]. However, it is unknown if oxidized residues, such as C36, in the NTD would have a similar CPVT-like pathology.

The hyper-active cysteine C822 is located in the SPRY1 (SPIa kinase and ryanodine receptor) domain, C917 in the tandem-repeat domain RY1&2, and C1582 in the SPRY3 region of RyR2. The SPRY domains represent important protein–protein interaction scaffolds in the periphery of the cytosolic shell of RyR. Interestingly, C1582 is located near the calstabin-binding site. Its oxidation can be important to modulating calstabin binding to the channel, which is critical for stabilizing the closed state of RyR and reducing spontaneous Ca^2+^ release. The RY1&2 domain is highly dynamic and its oxidation at C917 may contribute to the local regulation of this region.

The hyper-active cysteines C2402, C2572, C2577, and C3158 are located in the bridging solenoid (Bsol) of RyR2, another highly flexible and dynamic region of the protein. BSol encompasses the helical domains BSol1, BSol2, and BSol3. Between the first two is the RY3&4 domain that contains the PKA phosphorylation site (S2808) important for the adrenergic regulation of Ca^2+^ release by RyR2. Interestingly, helical domain 1 in the BSol1 domain is where C2402, C2572, and C2577 are located, and it exhibits a domain–domain interface with NTD. The disruption of NTD–Bsol interactions leads to inappropriate channel activation [251,252]. Similar to NTD, BSol1 represents another hotspot for pathogenic CPVT mutations, highlighting the important role of the NTD–Bsol interaction network in the allosteric modulation of the channel conformations. Residue C3158 is located in the BSol2 domain, which interacts with the RY1&2 and SPRY2 domains of the adjacent protomer. Figure 2Cysteine residues of ryanodine receptors prone to oxidation by ROS. (**A**) Evolutionary conservation of the hyper-active cysteines (red) across the RyR isoforms 1 and 2 in a rabbit, pig, mouse, and human. (**B**) Schematic diagram of the domain architecture of RyR2. Red circle in the diagram indicates the cysteines prone to oxidation for each RyR2 domain according to [243]. The same applies for the yellow circles, but according to [253] (**C**) Human RyR2 structure (PDB: 7UA3) showing the top and side views (**left**). The discussed oxidized cysteines are indicated in the close-up views (**right**). * means conserved residue among species.
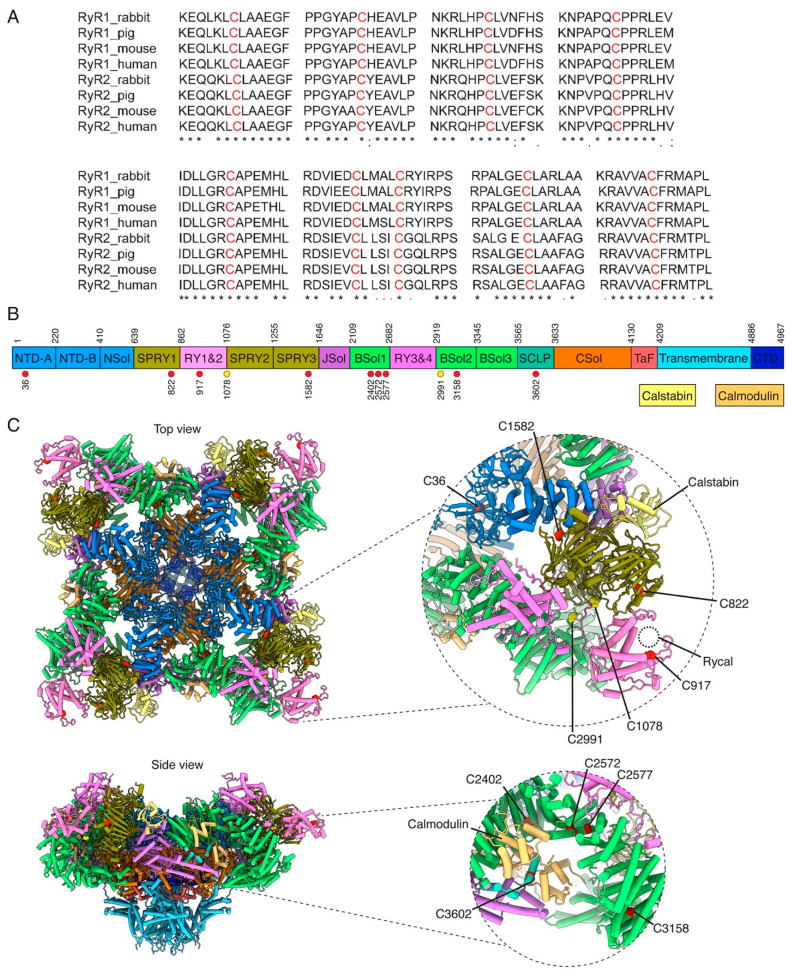



Nonetheless, through the application of genetic techniques to induce specific truncations and modifications in channel residues and domains, a recent study pinpointed that only two cysteine residues within each RyR2 subunit were accountable for nearly half of RyR2’s functional response when exposed to oxidative stress, specifically cysteine 1078 within the Ry1&2-SPRY2 linker and cysteine 2991 in the BSol loop function as a redox-sensitive pair. They engage in the formation of inter-subunit disulfide cross-links between adjacent subunits when confronted with oxidative stress, ultimately leading to an aberrant RyR2 Ca^2+^ leakage [253]. This finding aligns with the observation of elevated levels of RyR2 cross-linking in ventricular myocytes extracted from failing hearts [254].

Additionally, as previously discussed, while the previous research has pointed to cysteine 3635 in RyR1 or its equivalent, cysteine 3602, as the most hyperactive redox-sensitive residue, Nikolaienko et al. demonstrated that this particular residue did not contribute to channel cross-linking or the activation of Ca^2+^ leakage due to oxidative stress [245,253]. In concordance with previous discoveries, Nikolaienko et al. also showed that although none of the cysteine residues in the N-terminal domain (NTD) directly participated in cross-linking, they did play a pivotal role in allosterically supporting inter-subunit cross-linking within RyR2. Yet, none of the discussed residues were shown to be oxidized in human or animal models of HF. A further investigation into the specific mechanisms of RyR2 post-translational modification following oxidative stress, as well as the generation of cryo-EM structures with increasing resolution, is critical to understanding and designing therapeutic interventions for oxidative stress-mediated RyR2 pathological Ca^2+^ leak.

It has been observed that RyR channels self-organize into higher-order clusters forming checkerboard-like arrangements in the terminal cisternae of SR membranes. Peripheral domains likely involved in mediating clustering interactions between RyR channels are SPRY1, RY1&2 domains, and the C-terminal Bsol region [255,256]. Importantly, RyR clustering influences channel gating due to inter-RyR coupling [257,258]. Thus, we speculated that oxidation of hyper-active cysteines in these regions (C917, C822, and C3158) could modify RyR2 clustering and thereby its coordinated activity.

### 6.2. PKA and CamKII Oxidation and Modulation of SR Ca^2+^ Efflux

RyR2 is a large scaffold on which several regulatory proteins and enzymes can be assembled. Among them, two major regulatory kinases, protein kinase A (PKA) and Ca^2+^/calmodulin-dependent protein kinase II (CamKII), both known to be redox sensitive, have been shown to phosphorylate RyR2 at different sites and promote RyR2 Ca^2+^ leak. PKA and CamKII regulate RyR2 function and play an important role in regulating cardiac contractility and arrhythmogenesis [259,260,261,262,263,264,265,266,267,268,269]. Although it has been debated for decades, most studies lean toward the predominant role of the PKA-mediated phosphorylation of S2808 in HF. This post-translational modification depletes calstabin2 from RyR2 channels, causing Ca^2+^ leak, which in turn reduces cardiac contractility and ultimately furthers HF progression. Chronic PKA activation in HF is caused by the hyper-adrenergic state observed in HF patients, as well as in animal models of HF [4].

Furthermore, the type-I negative regulatory subunits of PKA are also subject to oxidation by ROS [270]. The oxidation of cysteines 17 and 38 leads to an inter-subunit disulfide bond formation between the two regulatory subunits and dissociation of the PKA holoenzyme complex. These processes allow PKA to phosphorylate several targets and increase cellular contractility without elevating 3′,5′-cyclic adenosine monophosphate (cAMP) levels [270], triggering pathological Ca^2+^ leak, which may lead to mitochondrial Ca^2+^ overload and dysfunction. To date, however, the exact physiological relevance of this ROS-dependent activation of PKA under physiological and pathological conditions, especially in comparison to adrenergic-dependent activation, remains mostly unknown.

Since CamKII has also demonstrated autonomous activation following the oxidation of its regulatory domain, the role of CamMKII oxidation in cardiovascular disease, including HF, has garnered significant attention [271]. The Anderson laboratory was one of the first to report that oxidized CamKII levels increased after myocardial infarction or angiotensin II injection, but not after isoproterenol [272]. They developed a panel of mutant CamKII to test the functional role of ROS-modifiable amino acids that were conserved across the four known CamKII isoforms. The authors suggested that CamKII could be activated by ROS as a consequence of M281/282 oxidation [272]. Animal models with a targeted loss of oxidized CamKII, in which M281/282 were replaced with valine, displayed decreased ROS induced-CamKII activation, reduced myocardial death, and improved left ventricular function after I/R injury. However, they retained other features of wild-type CamKII, including Ca^2+^/CaM autonomous activity due to T287 auto-phosphorylation [273]. Recently, Joiner and collaborators proposed that CamKII regulated mitochondrial Ca^2+^ uptake in the heart by phosphorylating the MCU, with subsequent impacts on mPTP activation and programmed cell death [274]. However, these findings were challenged by Fieni and coworkers, who did not detect significant effects of CamKII on MCU current density in experiments performed via the patch clamp of isolated mitochondria [275]. These data were consolidated using genetic approaches involving two mouse models with either the global deletion of CamKIIσ or the cardiomyocyte-specific deletion of CamKIIσ/ϒ, showing no significant effects on Ca^2+^ uptake, respiration, or ROS emission in isolated cardiac mitochondria, nor in isolated cardiac myocytes [276].

In addition to RyR2, PKA, and CamKII, SERCA2a has also been shown to experience dysregulation following oxidation by ROS. By treating cardiomyocytes with various reactive oxygen species, several studies have demonstrated decreased SERCA2a Ca^2+^ uptake into the SR, which may increase mitochondrial Ca^2+^ overload and further ROS production [277,278,279].

## 7. Mitochondrial Ca^2+^ Overload and Cell Death

Elevated mitochondrial matrix Ca^2+^ triggers the opening of the mitochondrial permeability transition pore (mPTP), a non-specific pore in the inner membrane of the mitochondria [280,281], converting them from organelles, whose production of ATP sustains the cell life, to instruments of cell death [282]. Thus, water and all solutes <1500 Da enter the mitochondrial matrix, leading to mitochondrial swelling, the unfolding of inner mitochondrial membrane cristae, and the disruption of the outer mitochondrial membrane, causing the release of cytochrome *c* and SMAC/Diablo into the cytosol and, eventually, apoptosis [282,283,284,285]. Pathological conditions that can open the mPTP include Ca^2+^ overload, production of ROS, depletion of ATP and ADP, and increases in inorganic phosphate [283], all of which are generally present during HF.

## 8. Therapies Targeting Ca^2+^ Defects/Mitochondrial Oxidative Stress in HF

As previously discussed, defective Ca^2+^ handling, mitochondrial dysfunction, and oxidative stress have been linked to the development and progression of HF, independent of etiology. Therefore, the SR and mitochondria may serve as two of the most promising therapeutic targets for HF [286,287,288,289,290,291]. Several drugs targeting mitochondria are currently under evaluation in clinical trials for HF patients. In light of these targeted therapies, an understanding of impaired mitochondrial Ca^2+^ handling, as well as its relation to altered cytosolic Ca^2+^, appears pivotal. Several strategies targeting mitochondria or the SR, including small molecules, peptides, and antioxidants that have been tested in preclinical and clinical studies in HF, are summarized in Table 1. We discuss in detail the properties and outcomes of two recent compounds that target the SR and mitochondria, Rycals and SS31, respectively.

### 8.1. Ca^2+^ Dysregulation-Targeted Therapies: Rycals

As mentioned above, RyR2 is an essential player in several functions of cardiomyocytes, from EC coupling to the activation of numerous pathways, gene expression, as well as mitochondrial Ca^2+^ overload and subsequent oxidative stress, thereby controlling the function and fate of cardiomyocytes. The impairment of RyR2 Ca^2+^ release has emerged as a major mechanism underlying HF and arrhythmogenesis as a consequence of genetic mutations and/or stress-mediated RyR2 post-translational modifications, loss of channel stability, and impairment of coupled gating.

A family of compounds known as Rycals, a small, orally available molecule, prevent RyR2 Ca^2+^ leak by preventing calstabin2 depletion without blocking the channel pore [246,247]. Promising results from the preclinical studies show the improvement of cardiac function in mice with dysfunctional RyR2. In rat and mouse models of post-myocardial infarction, RyR stabilizing compounds attenuated systolic and diastolic cardiac dysfunctions and preserved cardiac structure [292,293]. This finding was supported by recent experiments involving Rycal treatment of hiPS-derived cardiomyocytes obtained from skin fibroblasts or hair keratinocytes of patients with defective RyR2 mutations [283].

### 8.2. Mitochondria-Targeted Antioxidant SS31

SS31, a potent antioxidant peptide with the ability to penetrate cell membranes, is located within the inner mitochondrial membrane, where it exhibits protective properties for mitochondria [294,295]. Experimental studies conducted before the clinical trials have shown that SS31 provides protection in models of heart failure induced by pressure overload and hypertensive cardiomyopathy [186,296]. In the latter investigation, SS31 was found to prevent the accumulation of reactive oxygen species in mitochondria and alleviate angiotensin II-induced cardiac hypertrophy and diastolic dysfunction [294]. Furthermore, a newer version of SS31, known as Elamipretide (also referred to as Bendavia), was subjected to testing in two clinical trials involving a small group of patients [297]. One trial involved patients with heart failure characterized by a reduced ejection fraction (HFrEF, with 36 patients enrolled—NCT02388464) and the other focused on patients with an ST-segment elevation myocardial infarction (STEMI, with 300 patients enrolled—NCT01572909-EMBRACE) [298,299]. Both trials indicated acceptable safety and tolerability. However, the phase-2a trial EMBRACE did not demonstrate a significant improvement in the primary endpoint, which was myocardial infarct size reduction, following treatment. Subsequently, two modified versions of SS31, named mtCPP and mtgCPP, were ben developed, showcasing a 2–3-times-greater efficacy and antioxidant capacity compared to SS31 [300,301]. Nevertheless, whether these modified peptides possess the potential to emerge as new drug candidates for the treatment of heart failure in clinical settings remains to be described. Furthermore, despite the promising data and outcomes provided by SS31, there are no studies showing its effect on cardiac Ca^2+^ homeostasis, known as the major mechanism in cardiomyopathies, arrythmias, and death.biomolecules-13-01409-t001_Table 1Table 1Oxidative stress/Ca^2+^ dysregulation-targeted therapies in cardiac disease.AgentTargetActionDiseasesDose/SpeciesDevelopmentOutcomesReferencesAAV1/SERCA2aSERCA2aSERCA2a overexpressionHeart failureIschemic cardiomyopathyPatients1 × 10^11^–1 × 10^13^ DNase particlesTerminated clinical trialsNo effect/reduced cardiac events 1 year after[277,302]NCT00534703NCT01966887NCT02346422Alda-1MitoALDH2Increased ALDH2 activityI/R injuryHF post-MIRat 8.5 mg/kgPre-clinicalReduced infarct size by 60%[303,304,305,306]Cyclosporine AMTPMTP inhibitionI/R injuryHF post-MIPatients 2.5 mg/KgIVPhase 3Reduced infarct size[307,308,309]EUK-8SODSOD/catalase mimeticDCMPressure overload-induced HFMice 30 mg/kg/dayIPPre-clinicalPrevented DCM in miceAmeliorated systolic function and survival[310,311]IdebenoneCoenzyme Q10Free radical scavengerMitochondrial cardiomyopathy225 mg/dayIn patientsPre-clinical phase 3Increased EF by >50%(a case report)[312]M40403SODSOD/catalase mimeticI/R heart injuryRats 1 to 10 mg/kgIVPre-clinicalProtected tissue damage after I/R in rats[313,314]mitoTEMPOMitochondrialnitroxideROS scavengerDiabetic cardiomyopathyHypertensionMice 0.7 mg/kg/dayIPPre-clinicalReduced myocardial hypertrophy[315,316,317]MCI-186(Edaravone)Free radicalsFree radical scavengerHFAcute ischemic strokepatients30 mgIVPre-clinical phases 2–4Reduced enzymatic infarcts/better clinical outcomes[318,319]MitoQETCFree radical scavengerPressure overloadCardiovascular functionMice 100 uM/DWRatsUnavailable for patientsPre-clinical2 clinical trialsDecreased heart dysfunction[320,321]NCT03506633NCT03586414MetforminETCETC inhibitionI/R injuryHF post-MIHFpEFMice and rats 200–250 mg/kgPhase 2Improved cardiac function (rats and mice)[322,323]NCT03629340Rycals (S107-ARM210)Ryanodine receptor (RyR)Stabilizing RyRI/R injuryHF post-MIMice 20–75 mg/kg oralPreclinical studyImproved cardiac function/reduced arrhythmias[324]NCT04141670SS31MitochondriaCardiolipin protectionHFrEF, HFpEFCongestive HFPatients 0.25 mg/kg/hClinical phases 1 and 2Improved cardiac volumes[298]NCT02814097NCT02914665TRO-40303MTPMTP inhibitionI/R injuryHF post MIPatients 2.5 mg/kgPhase 2No effectReduced infarct size by 38%[325,326,327]XJB-5-131MitochondrialnitroxideROS scavengerI/R heart injuryRats 3 mg/KgIPPre-clinicalImproved post-ischemic recovery[278,279]


## 9. Conclusions

Beyond their crucial role as an energy source, cardiac mitochondria serve as Ca^2+^ reservoirs. They accumulate Ca^2+^ ions during cytosolic Ca^2+^ elevations in cardiomyocytes. Elevations in intramitochondrial Ca^2+^ levels trigger multiple enzymes within the mitochondrial matrix. These enzymes, in turn, boost ATP generation, modify the patterns of intracellular Ca^2+^ signaling across both space and time, and assume a pivotal role in instigating mitochondrial oxidative stress and pathways leading to cell death. It is clear that, in HF, an excessive SR Ca^2+^ leak, mainly through RyR2 channels, plays a crucial role in the pathophysiology of mitochondrial Ca^2+^ accumulation and dysfunction and constitutes a feedback loop of alterations between the SR and mitochondria, which contribute to the impairment of cardiac function. Future investigations should focus on breaking this feedback loop by simultaneously targeting the SR leak and mitochondrial Ca^2+^ overload/oxidative stress using RyR2 stabilizing drugs and mitochondrial antioxidants.

## Figures and Tables

**Figure 1 biomolecules-13-01409-f001:**
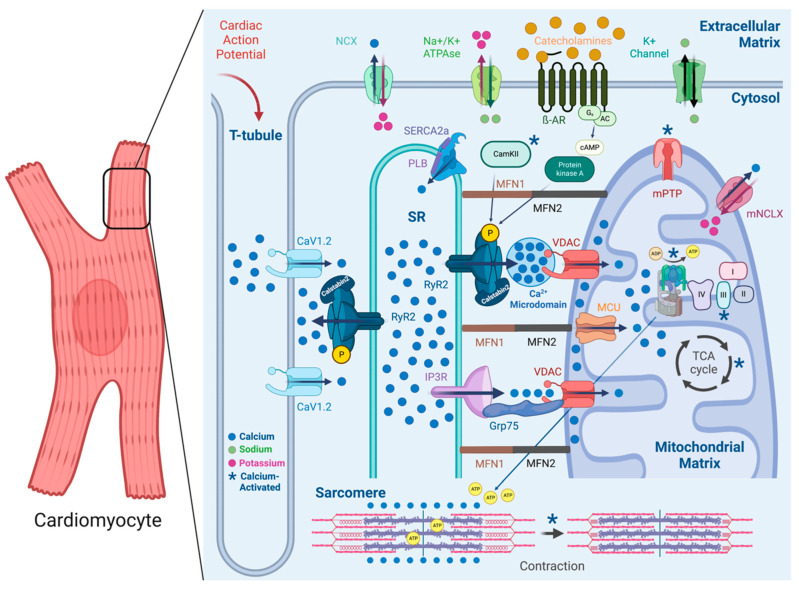
Physical interaction and Ca^2+^ transfer from the SR to the mitochondria in cardiomyocytes: during the depolarization phase of the cardiac action potential, Ca^2+^ enters the cardiomyocyte through voltage-activated L-type Ca^2+^ channels (CaV1.2) and triggers RyR2 to open and release Ca^2+^ from SR Ca^2+^ stores through calcium-induced calcium release (CICR). RyR2 activity can be increased by CaMKII or PKA phosphorylation, in response to stress. Ca^2+^ release from the SR by RyR2 creates high [Ca^2+^]_cyt_ microdomains for mitochondrial Ca^2+^ uptake. Close contact points between the SR and mitochondria are controlled by SR mitofusin 2 (MFN2) tethering to MFN2 and MFN1 on the outer mitochondrial membrane (OMM), further creating microdomains of a high Ca^2+^ concentration in the vicinity of mitochondria. Ca^2+^ release from the SR by IP3R transits through the protein complex formed by IP3R/GRP75/VDAC and enters the mitochondria through the MCU, later being removed by mNCLX. Ca^2+^ activates complex III of the electron transport chain, ATP Synthase, several dehydrogenases in the TCA cycle, mPTP, CaMKII, and contraction of the sarcomeres. To induce repolarization, SERCA2a and NCX cycle Ca^2+^ out of the cytosol.

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
