# Peer review of "Mitochondrial Calcium Overload Plays a Causal Role in Oxidative Stress in the Failing Heart"

_biomolecules, 2023, doi:10.3390/biom13091409_

Round 1
Reviewer 1 Report
Thank you for the opportunity to review this manuscript entitled ‘Mitochondrial calcium overload plays a causal role in oxidative stress in the failing heart’ by Haikel Dridi and co-authors.
The present study dedicated to finding the mechanisms of mitochondrial Ca2+ uptake and the molecular pathways involved in the regulation of increased mitochondrial Ca2+ influx that is the main challenge in order to identify novel therapeutic targets to heart failure.
This problem is very relevant due to the cardiovascular diseases remain the leading cause of death worldwide. Search for molecular mechanisms and new treatment strategies for patients with heart failure is critical to improving their quality of life. This review is in line with the work of the team of authors. The manuscript is well written and logical and presents review of current scientific evidence appropriate for publication in Biomolecules journal in present form.
Author Response
Reviewer #1: Thank you for the opportunity to review this manuscript entitled ‘Mitochondrial calcium overload plays a causal role in oxidative stress in the failing heart’ by Haikel Dridi and co-authors.
The present study dedicated to finding the mechanisms of mitochondrial Ca2+ uptake and the molecular pathways involved in the regulation of increased mitochondrial Ca2+ influx that is the main challenge in order to identify novel therapeutic targets to heart failure. This problem is very relevant due to the cardiovascular diseases remain the leading cause of death worldwide. Search for molecular mechanisms and new treatment strategies for patients with heart failure is critical to improving their quality of life. This review is in line with the work of the team of authors. The manuscript is well written and logical and presents review of current scientific evidence appropriate for publication in Biomolecules journal in present form.
Response: we thank the Reviewer for their positive comments on our manuscript.
Reviewer 2 Report
This is a comprehensive review, and will make a useful contribution to the literature.
Some questions or concerns I had follow:
1. "Mitochondrial calcium overload" is used throughout, and its importance indicated by the title of the Review. However, I am still left wondering what mitochondrial "overload" is exactly.
I'd have liked to see some examples of mitochondrial calcium fluxes in intact cells. In our experience, mitochondria take up and release calcium within one cardiac cycle. With mitochondria making up ~30% of cell volume, what mitochondrial calcium concentration constitutes "overload"? We know at some stage the opening of the MTP occurs, which is bad news for the myocytes. But before then? Is there some upper limit for mitochondrial calcium accumulation that does not permanently damage the mitochondria? I ask this because in my work with isolated cardiac muscle preparations it is possible to observe cytosolic calcium concentration to increase (e.g. during high [K+] contractures) to many times the peak calcium transient concentration and return again to apparently healthy cellular calcium handling on K+ washout. At other times, irreversible damage to the cells occurs with much lower Ca2+ conc exposure. We routinely work with human atrial tissue, some of which has extensive calcium leak during diastole, yet these preparations still respond to high frequency stimulation and isoproterenol and recover without irreversible damage.
So, I think "mitochondrial calcium overload" is perhaps more subtle than just an upper limit for mitochondrial calcium accumulation.
2. Unlike many other cell types, cardiac myocytes shorten and lengthen during each cardiac cycle. What impact does this have on structures tethering/linking SR to mitochondria? There must be some considerable flexibilty, for when cytosolic Ca2+ is maximal the sarcomeres are contracted.
RyR2 are mainly located at the dyads, whereas SERCA2a is distributed along the non-junctional SR. Is it reasonable to think a very high Ca2+ during SR release can rapidly increase ATP production sufficiently to match the increased demand of SERCA activity?
3. I thought Section 5 was too long. To my mind the content could be reduced. It would perhaps be the focus of a separate review article.
4. The conclusion was very brief. Should we be targeting mitochondrial calcium uptake, or should we be more concerned about ROS and finding ways to protect against the damage they cause?
Mostly the English was fine. However, there were a few instances of misuse of words (e.g. "evolutionary" instead of "evolutionarily") and grammatical errors (singular instead of plural etc).
There needs to be someone with more time than I have to list the corrections that should be made.
Author Response
Reviewer #2:
This is a comprehensive review and will make a useful contribution to the literature.
Response: We thank the Reviewer for their positive comments on our manuscript.
"Mitochondrial calcium overload" is used throughout, and its importance indicated by the title of the Review. However, I am still left wondering what mitochondrial "overload" is exactly. I'd have liked to see some examples of mitochondrial calcium fluxes in intact cells. In our experience, mitochondria take up and release calcium within one cardiac cycle. With mitochondria making up ~30% of cell volume, what mitochondrial calcium concentration constitutes "overload"? We know at some stage the opening of the MTP occurs, which is bad news for the myocytes. But before then? Is there some upper limit for mitochondrial calcium accumulation that does not permanently damage the mitochondria? I ask this because in my work with isolated cardiac muscle preparations it is possible to observe cytosolic calcium concentration to increase (e.g. during high [K+] contractures) to many times the peak calcium transient concentration and return again to apparently healthy cellular calcium handling on K+ washout. At other times, irreversible damage to the cells occurs with much lower Ca2+ conc exposure. We routinely work with human atrial tissue, some of which has extensive calcium leak during diastole, yet these preparations still respond to high frequency stimulation and isoproterenol and recover without irreversible damage.
So, I think "mitochondrial calcium overload" is perhaps more subtle than just an upper limit for mitochondrial calcium accumulation.
Response: We thank the Reviewer for their very relevant comments. The precise quantity of Ca2+ absorbed by mitochondria on a beat-to-beat basis in both normal and failing hearts is not clearly understood. There is a lack of consensus regarding whether mitochondria play a significant role as a storage site for Ca2+. There have been limited investigations directly focused on examining how mitochondrial Ca2+ dynamics are affected in cases of heart failure. The majority of evidence indicating potential Ca2+ harm in mitochondria arises from experiments that manipulate intracellular Ca2+ balance using genetic methods. Notably, excessive cytosolic Ca2+ accumulation in various animal models results in mitochondrial behaviors resembling those observed in heart failure, such as the opening of mPTP, heightened mitochondrial oxidative stress, disruption of mitochondrial membrane potential, impaired ATP generation, and the death of heart muscle cells through necrosis. Whether a comparable degree of mitochondrial Ca2+ overload takes place in heart failure and the mechanisms behind it remain subjects requiring further investigation.
- Unlike many other cell types, cardiac myocytes shorten and lengthen during each cardiac cycle. What impact does this have on structures tethering/linking SR to mitochondria? There must be some considerable flexibilty, for when cytosolic Ca2+ is maximal the sarcomeres are contracted.
Response: There is no evidence that mitochondria move or change tethering/linking to the SR during cardiac cycle. Mitochondria dynamics are driven mainly by the energy demand and or changes in the cell homeostasis including changes in the cytosolic Ca2+ concentration.
RyR2 are mainly located at the dyads, whereas SERCA2a is distributed along the non-junctional SR. Is it reasonable to think a very high Ca2+ during SR release can rapidly increase ATP production sufficiently to match the increased demand of SERCA activity?
Response: Although this remains hypothetical, we agree with the Reviewer that it is very reasonable to think that the very high cytosolic Ca2+ during the systole, would first contribute to the cardiomyocytes shortening but also would stimulate the mitochondrial ATP production which is necessary for SERCA activation in order to pump the Ca2+ back into the SR and prime the cardiac relaxation.
- I thought Section 5 was too long. To my mind the content could be reduced. It would perhaps be the focus of a separate review article.
Response: Section 5 focuses on the potential RyR2-redox sensitive residues which may be subjected to oxidation and worsening the SR Ca2+ leak in HF. There is substantial controversy in the literature about which residue is functionally relevant. In the recent years, our lab has gained more insights into the human RyR2 structure and dynamics and we wanted to discuss the literature from a protein structure perspective. We believe that this section is the key novelty of this review and addresses how oxidative stress increases SR Ca2+ leak. We have revised this section and made it shorter and more cohesive.
- The conclusion was very brief. Should we be targeting mitochondrial calcium uptake, or should we be more concerned about ROS and finding ways to protect against the damage they cause?
Response: We believe that there is no conclusive answer to this question. Mitochondrial Ca2+ overload and oxidative are inextricably connected. Indeed, mitochondrial Ca2+ overload leads to oxidative stress and oxidative stress damages the Ca2+ regulating proteins which cause more mitochondrial Ca2+ overload and lead to a vicious cycle of alterations. In HF, an early neurohormonal signaling (The catecholaminergic response) initiate the alteration of RyR2 through a PKA hyperphosphorylation, leads to cytosolic Ca2+ overload and mitochondrial accumulation. Rycal drugs or genetic manipulation which prevent RyR2 defects have shown a beneficial effect on mitochondrial structure and function and the cardiac function. Mitochondrial antioxidant treatment has also shown beneficial effects on the cardiac contractility. We believe that future studies should test a bitherapeutic intervention to simultaneously reduce SR Ca2+ leak and mitochondria ROS production. We added this point of view to the revised conclusion.
Comments on the Quality of English Language
Mostly the English was fine. However, there were a few instances of misuse of words (e.g. "evolutionary" instead of "evolutionarily") and grammatical errors (singular instead of plural etc). There needs to be someone with more time than I have to list the corrections that should be made.
Response: We have revised the grammatical errors.
Reviewer 3 Report
This review focus on alterations in cardiomyocyte Ca2+ handling, involving mitochondria, in heart failure independent on its aetiology. Heart failure is clinical syndrome and dynamic process resulting from cardiovascular disorders of various aetiology that finally aggravates to heart contractile dysfunction. The alterations in Ca2+ handling included in the review should be related to the various degree (stages) of heart failure. Moreover, it should be taken into consideration that there are clinically relevant stages of heart failure that are treated and may affect Ca2+ handling as well. The topic of the review is actual from both experimental and clinical point of view, however, some issues should be taken into consideration during revision of the discussed data. The authors should include most relevant references dealing with Ca2+ handling in failing heart. Some, references not related to the heart or heart failure, e.g. 35 etc., or those not helping to solve the discussed issue should be reduced.
Tile: Should reader understand that mitochondrial calcium overload promotes oxidative stress in failing heart? If so, exact data should be presented in the manuscript to support this view.
Abstract: Instead statistic, the definition of heart failure is missing. The information as whether reviewed data dealing with Ca2+ handling is mostly from experimental models of heart failure, is missing. What was the criteria, according them heart failure was defined and respective Ca2+ handling analysed? Conclusion should be more precise, in which stadium of heart failure therapeutic strategies are suggested.
Introduction: Most recent papers dealing with heart failure as should be cited instead old one (ref 1-5), e.g. Da Dalt 2023, Li Ng 2023). The information as whether reviewed data dealing with Ca2+ handling is mostly from experimental models of heart failure, is missing.
„The less is more”. Keep as much as possible the main point of the review with citations of the most relevant data and avoid subsidiary information (including references). If possible characterize the stage of heart failure (recent, chronic, or according to NYHA classification) in which published data were acquired. Avoid misinterpretations as f.e. on page 14, that many different studies to cause mitochondrial ROS production in several diseases, including HF (52, 69). There are old references dealing with drugs targeting mitochondria in heart failure and not recent) see page 23.
Therapeutic approaches to attenuate mitochondrial Ca2+ overload should be more exactly characterized based on recent papers.
Recommendations for further studies or clinical examinations may be appreciated.
Conclusions: It should be concluded whether and how harmful Ca2+ signalling in mitochondria may be attenuated or prevented in failing heart.
Author Response
Reviewer #3:
This review focus on alterations in cardiomyocyte Ca2+ handling, involving mitochondria, in heart failure independent on its etiology. Heart failure is clinical syndrome and dynamic process resulting from cardiovascular disorders of various etiology that finally aggravates to heart contractile dysfunction. The alterations in Ca2+ handling included in the review should be related to the various degree (stages) of heart failure. Moreover, it should be taken into consideration that there are clinically relevant stages of heart failure that are treated and may affect Ca2+ handling as well. The topic of the review is actual from both experimental and clinical point of view; however, some issues should be taken into consideration during revision of the discussed data. The authors should include most relevant references dealing with Ca2+ handling in failing heart. Some, references not related to the heart or heart failure, e.g. 35 etc., or those not helping to solve the discussed issue should be reduced.
Response: We thank the Reviewer for their positive comments on our manuscript. We have added the most recent references related to heart failure.
Tile: Should reader understand that mitochondrial calcium overload promotes oxidative stress in failing heart? If so, exact data should be presented in the manuscript to support this view.
Response: Our laboratory has shown that that diastolic SR Ca2+ leak, through leaky RyR2, causes mitochondrial Ca2+ overload and dysfunction in a murine model of post-myocardial infarction HF (PMID: 26217001). This mitochondrial Ca2+ overload was correlated with enhanced mitochondrial ROS production and oxidative stress. We showed that RyR2 mutations that cause SR Ca2+ leak, result in mitochondrial Ca2+ overload, oxidative stress, dysmorphology, and dysfunction. Interestingly, when the SR Ca2+ leak was prevented by genetic manipulation or by stabilizing RyR2 channels using Rycals, mitochondrial Ca2+ content was significantly reduced and oxidative stress was attenuated. Furthermore, we have shown that genetic enhancement of mitochondrial antioxidant activity by over-expressing mitochondrial catalase improved mitochondrial function, reduced posttranslational modifications of RyR2 macromolecular complex and improved heart function. All these data strongly support the role of mitochondrial Ca2+ overload in mediating oxidative stress in HF. We have discussed these data in the revised version of the manuscript.
Abstract: Instead statistic, the definition of heart failure is missing. The information as whether reviewed data dealing with Ca2+ handling is mostly from experimental models of heart failure, is missing. What was the criteria, according to them heart failure was defined and respective Ca2+ handling analyzed? Conclusion should be more precise, in which stadium of heart failure therapeutic strategies are suggested.
Response: We added the definition of HF to the introduction of the revised manuscript. This review focuses on the role of mitochondrial/SR molecular mechanisms involved in HF. Most of the discussed studies were performed in animal model of heart failure with reduced ejection fraction. These models often exhibit a structurally abnormal heart (ischemic area, dilated and hypertrophic heart, increased fibrosis), reduced ejection fraction below 35%, reduced exercise capacity and muscle dysfunction. All these features are characteristics of decompensated HF and would correspond to stage C2 and/or D in HF patients according to the NYHA classification.
Introduction: Most recent papers dealing with heart failure as should be cited instead old one (ref 1-5), e.g. Da Dalt 2023, Li Ng 2023). The information as whether reviewed data dealing with Ca2+ handling is mostly from experimental models of heart failure, is missing.
Response: The review by Da Dalt et al 2023 discussed the lipid metabolism in heart failure which is out of scope in the current review. We were not able to find the Li Ng 2023 reference.
“The less is more”. Keep as much as possible the main point of the review with citations of the most relevant data and avoid subsidiary information (including references). If possible, characterize the stage of heart failure (recent, chronic, or according to NYHA classification) in which published data were acquired. Avoid misinterpretations as f.e. on page 14, that many different studies to cause mitochondrial ROS production in several diseases, including HF (52, 69). There are old references dealing with drugs targeting mitochondria in heart failure and not recent) see page 23.
Response: We removed some of the old citations in the revised version of the manuscript.
Therapeutic approaches to attenuate mitochondrial Ca2+ overload should be more exactly characterized based on recent papers.
Response: The current review focuses on the SR/mitochondria interaction/signaling through Ca2+ and oxidative stress. The list of therapeutic interventions is long, mainly for antioxidant therapeutics. We summarized these therapeutics in table 2. We have included only the most advanced therapeutic intervention for each organelle (Rycals for the SR Ca2+ leak, SS31 for the mitochondrial oxidative stress) to avoid redundancy with previously published reviews. We added and discussed the recent publication by Garbincius et al JMCC 2022 and discussed their findings on mitochondrial NCLX overexpression in mice.
Recommendations for further studies or clinical examinations may be appreciated.
Response: We have added this to the conclusion.
Conclusions: It should be concluded whether and how harmful Ca2+ signaling in mitochondria may be attenuated or prevented in failing heart.
Response: We have added this to the conclusion.
Round 2
Reviewer 3 Report
Thanks for your responses to my comments.